# Antiviral Drugs in Adenovirus-Induced Keratoconjunctivitis

**DOI:** 10.3390/microorganisms10102014

**Published:** 2022-10-12

**Authors:** Roberto Imparato, Nicola Rosa, Maddalena De Bernardo

**Affiliations:** Eye Unit, Department of Medicine, Surgery and Dentistry, Scuola Medica Salernitana, University of Salerno, 84081 Salerno, Italy

**Keywords:** adenovirus, conjunctivitis, antiviral drugs, keratitis

## Abstract

Human adenovirus (HAdV) is one of the most common causes of conjunctivitis worldwide. Depending on specific serotypes and other factors, it can lead to several ocular manifestations, ranging from isolated, self-limited disease to epidemic and potentially sight-threatening keratoconjunctivitis. To date, no antiviral agent against ocular adenovirus has been licensed, and its management is still based on hygienic and supportive measures alone. In this review, a literature search up to August 2021 was performed to find peer-reviewed articles, with the primary aim to investigate drugs or other compounds with any antiviral activity against adenovirus. Finally, we included 70 articles, consisting of both in vitro, and in vivo studies on animal models and clinical trials of any phase, as well as a case-report, and analyzed each compound separately. Many antiviral agents proved to be effective on in vivo and in vitro studies on animal models, and in pre-clinical trials, but lacked reliability in large, controlled clinical investigations. The design of such studies, though, presented several hurdles, due to the nature and the specific characteristics of adenovirus-induced ocular diseases. Nevertheless, some promising compounds are currently under study, and further investigations are needed to prove their efficacy in the management of adenovirus conjunctivitis.

## 1. Introduction

Human adenovirus (HAdV) is a ubiquitous virus that can infect different types of mucoepithelial cells of the human body. Therefore eyes, as well as genitourinary, respiratory, and gastrointestinal tracts can be considered primary targets of HAdV infection in humans [1].

Conjunctivitis is one of the most frequent conditions seen in the eye clinics all over the world [2]. It can be caused by several pathogenic agents, although a viral infection is the most common [3].

Among all the viruses, HAdV represents the most common cause of conjunctivitis. The spectrum of disease severity varies with serotypes: while serotypes 1–11 and 19 mainly cause a follicular conjunctivitis, serotypes 3, 4, 5, and 7 are responsible agents of pharyngoconjunctival fever; serotypes 3, 4, 8, 11, 19, and 37 produce acute conjunctivitis; and types 8, 9, and 37 are responsible for epidemic keratoconjunctivitis [4].

The epidemic keratoconjunctivitis, followed by pharyngoconjunctival fever, is the most common manifestation of ocular HAdV infection. Watery eyes, hyperemia, chemosis, and lymphadenopathy are the most common signs and symptoms of epidemic keratoconjunctivitis. Rapid onset of fever, pharyngitis, conjunctivitis, and periauricular lymph node enlargement are characteristics of pharyngoconjunctival fever. Corneal or systemic involvement are not always described [5].

The diagnosis of adenoviral conjunctivitis is usually obtained through clinical examination, and its differential diagnosis includes other causes of “red eye” (such as uveitis, glaucoma, scleritis and traumas). Nevertheless, nowadays an increasing number of laboratory tests to rule out adenoviral conjunctivitis is available.

This laboratory-supported early identification can assist physicians to start proper hygienic precautions. The available assays include those using cell culture, electron microscopy, or antigen or nucleic acid detection. Though less sensitive and specific than tests for nucleic acid detection, commercially available quick tests for the viral antigen detection are nonetheless helpful, as they can be quickly and readily carried out in clinical practice [6].

Although adenoviral conjunctivitis is mostly a benign, self-limited condition, visual disability is a possible outcome; therefore, an early diagnosis and an appropriate treatment could be mandatory. A known or authorized medication, specifically effective against adenovirus, is not yet available. Adjunctive care mainly consists of cold compresses and artificial tears.

Although the use of topical steroids may be useful in the treatment of ocular inflammation, especially considering tissue-scarring prevention, it is reserved for cases of severe conjunctivitis in most of the guidelines. This is partly due to the risks associated with long-term treatments with such drugs [5]. It is only where a bacterial coinfection is suspected, or in high-risk individuals, that topical treatment with antibiotics is suggested [7]. Concerning antiviral drugs to treat HAdV’s conjunctivitis, as their efficacy is controversial and many of them are characterized by severe adverse effects, none of them has been approved [8].

## 2. Methods

The aim of this review was to search for medical literature concerning the use of antiviral drugs to treat adenoviral conjunctivitis. Keywords used in PubMed to perform our search for articles published up to August 2021 were “adenoviral conjunctivitis” and “antiviral”. References of relevant articles, as well as reviews on this topic, were also searched for manually. 

We included in the present review only peer-reviewed articles whose primary aim was to investigate drugs or other compounds with any antiviral activity against adenovirus. We included both in vitro and in vivo studies on animal models and clinical trials of any phase, as well as a case-report, and divided them, therefore, into different sections, when needed. 

We excluded literature written in languages other than English. Articles investigating treatments for adenoviral conjunctivitis that did not include any antiviral drug or compound with primary activity against adenovirus were also excluded.

A total of 70 articles were included in this review. Among these articles, we were able to find 5 articles dealing with ganciclovir, 11 dealing with cidofovir, 8 dealing with ribavarine, 8 with zalcitabine, 3 with vidarabine, 2 with idoxuridine, 5 with trifluridine, 1 with acyclovir, 8 with filociclovir, and 19 with other agents.

## 3. Results

### 3.1. Some of These Compounds Were Tested Only in Laboratory Studies

#### 3.1.1. Ribavirin

Ribavirin is a guanosine analogue with a broad variety of antiviral activity, including flu viruses, RSV, and para-flu viruses. Ribavirin is quickly phosphorylated by intracellular enzymes and the triphosphate inhibits flu virus RNA polymerase activity and competitively prevents the guanosine triphosphate–dependent 5′ capping of flu viral messenger RNA. Moreover, ribavirin reduces cellular guanine pools and may prevent virus replication by lethal mutagenesis [9] and is effective for curing life-threatening systemic adenoviral infections [10].

##### Laboratory Studies

Two major problems were revealed while investigating potential efficacy of ribavarine on adenovirus infections in both in vitro and in vivo studies on animal models: first, the low clearing capacity on adenovirus-infected cells [11,12,13,14]; second, the lack of efficacy on serotypes 8, 19, and 37 in vitro [10,15,16,17] as well as an IC50 that was considered resistant for serotypes belonging to species A, B, D and F [18].

Considering the unpromising results in preclinical studies, and the above-mentioned adenovirus serotypes, that represent a major source of infection in human eyes, any further investigation on potential benefit of ribavirin against ocular adenovirus was dismissed.

#### 3.1.2. Zalcitabine

Zalcitabine is a pyrimidine analog derived from cytidine. The Food and Drug Administration (FDA) authorized it in 1992 for use in conjunction with zidovudine (ZDV) in adult HIV-infected patients who had clinically or immunologically worsening. Two antiretroviral mechanisms have been proposed: integration of dideoxynucleoside triphosphates onto expanding strands of viral DNA or competition for reverse transcriptase with endogenous nucleoside triphosphates [19]. 

##### Laboratory Studies

Zalcitabine showed efficacy against multiple adenovirus serotypes in vitro [20,21,22,23,24] and is active against numerous adenovirus serotypes, with IC50 values ranging from 3.5–48.7 mg/mL [25]. Additionally, the HIV-effective antiviral 6-azacytidine seems to reduce adenovirus replication in vitro [26,27,28,29,30,31,32,33,34,35,36].

### 3.2. Other Compounds Showed a Poor Efficacy

#### 3.2.1. Acyclovir

Acyclovir carries out its antiviral action inhibiting herpesvirus DNA polymerase. It is effective against *Herpes simplex* and varicella-zoster viruses in vitro. As for its mechanism of action, acyclovir triphosphate competitively inhibits viral DNA polymerase, integrates into, and ends the developing viral DNA chain, and inactivates the viral DNA polymerase. The drug may be administered topically to the skin, intravenously, orally, or topically to the eye. Acyclovir and its metabolites are eliminated by the kidney via glomerular filtration and tubular secretion [37]. 

##### Clinical Studies

Considering the treatment of adenoviral conjunctivitis with topical acyclovir, we found a case report [38] of a 17-year-old man treated with topical acyclovir ointment 3% four times per day and topical corticosteroids and antibacterial drugs for a suspected HSV-induced keratoconjunctivitis. Acyclovir administration was dismissed when a coexistent adenoviral and *Acanthamoeba* keratoconjunctivitis was finally diagnosed.

#### 3.2.2. Vidarabine

Vidarabine is a nucleoside analogue that was once mainly used as anticancer agent thanks to its property to interfere with tumor growth by inhibiting DNA replication [28]. Upon cellular uptake, this nucleoside analogue is physiologically triggered by 5′-triphosphorylation. Then, it inhibits intracellular enzymes and/or delays or terminates nucleic acid production, as they are integrated into nascent DNA and RNA strands [29,30]. Vidarabine, though no more employed as anti-tumoral therapy, because of its rapid deamination in vivo, is nevertheless helpful in the treatment of *Herpes simplex* and varicella zoster virus diseases [31].

##### Laboratory Studies

In vitro and in vivo studies, as well as clinical trials, showed that vidarabine and other virustatic agents are unsuccessful or slightly helpful to treat ocular adenoviral diseases [5,32].

##### Clinical Studies

Researchers in one trial [33] compared vidarabine treatment to a placebo in the prevention of subepithelial corneal infiltrates, during an outbreak of epidemic keratoconjunctivitis caused by adenovirus types 3, 7, 8, and 19. Subepithelial infiltrates were employed as a criterion for antiviral effectiveness because they are a clinical sign that is easily identified, whereas keratoconjunctivitis itself is very variable and self-limiting. A total of 36 patients out of 72 received 3.3% vidarabine in topical administration, while 42 received polyvinyl alcohol eyedrops, both every hour during daytime. Vidarabine was found to be ineffective in preventing subepithelial corneal infiltrates in adenovirus-induced keratoconjunctivitis. In fact, corneal infiltrates were noticed in 81% of patients who had received vidarabine, compared to 63% of control subjects, by an observer, who was unaware of the prescribed medication. 

#### 3.2.3. Idoxuridine

Thymidine analogue idoxuridine (IDU) is also an effective reversible inhibitor of the enzyme thymidilate synthetase, which converts irridine monophosphate to thymidine monophosphate. The DNA synthesis is also decreased because of this thymidilate synthetase inhibition [34].

##### Clinical Studies

Hecht et al. [35] recruited 18 patients diagnosed with type 8 Adenovirus during a small outbreak of epidemic keratoconjunctivitis. A first group of patients consisted of seven patients who were put on the double-blind schedule and two on known idoxuridine. A second group of patients was made up of nine who were not referred to the study but were followed either in the clinic or were inpatients. Two of these cases were treated with known idoxuridine. As for the therapeutic regimen, the patients were treated either with sterile distilled water and 0.002% thimerosal, or 0.1% idoxuridine, as well as with local antibiotics. Therapy was continued for two to three weeks. Several criteria were chosen to establish whether antiviral treatment was effective, such us length of acute phase, conjunctival disease, as well as onset and severity of corneal involvement; lastly, idoxuridine had no appreciable beneficial effect on the disease history when therapy was initiated at the beginning of corneal involvement. Despite treatment, new corneal lesions tended to grow in number in all cases.

A double-blinded, randomized clinical trial [36] tried to establish if IDU applied topically as ointment was effective in treating epidemic keratoconjunctivitis associated with adenovirus infection. Among 35 patients with virologically tested adenoviral infection, 17 were put in therapy with IDU and 18 were followed as control group. The early clinical severity of infection was similar between two groups—only unilateral cases were considered, and keratitis was not present when the treatment started. Subjects were treated as outpatients either with 0.5% IDU made up in ointment form or with the ointment base alone. With this regimen no beneficial effect of IDU was detected in patients with adenovirus infections of the eye. Despite the institution of treatment early in the course of the disease there was no evidence that the drug produced any reduction in the prevalence or severity of the keratitis. The adenovirus infections appeared to run their natural course uninfluenced by the presence of IDU.

### 3.3. The Following Drugs Seem to Show Some Degree of Efficacy

#### 3.3.1. Ganciclovir

Ganciclovir (9-[(1,3-dihydroxy-2-propoxy)methyl]guanine), an Acyclovir derivate, is a powerful inhibitor of herpesviruses. The viral DNA replication suppression by ganciclovir-5’-triphosphate (ganciclovir-TP) is its principal mechanism. This involves a strong and specific viral DNA polymerase inhibition [11]. Ganciclovir is approved in both the United States and the European Union for the hCMV infections therapy in immunocompromised patients, and it is also approved in some European countries for the herpes-induced keratitis topical treatment [11]. 

##### Laboratory Studies

Some in vitro studies, as well as investigations on animal models initially suggested a beneficial activity of the drug on common ocular adenovirus serotypes [39]. An in vitro study [40] investigated the efficacy of ganciclovir on some specific adenovirus serotypes (in particular, 3, 4, 8, 19a and 37), by using serial dilutions of ganciclovir over 24 h and quantitatively measuring adenoviral DNA by real-time polymerase chain reaction (PCR). Ganciclovir proved to significantly inhibit the replication of all the serotypes. Another in vitro study [41] employed A549 cells for viral cell culture, using the same adenovirus types. In this case, adenovirus was cultured for 7 days after pretreatment with sequential dilutions of ganciclovir for 24 h, and adenoviral DNA was quantified using real-time PCR. If ganciclovir can help to reduce viral loads in vitro, it could show some efficacy on animal ocular models. Researchers studied ganciclovir’s effectiveness in immunosuppressed Syrian hamsters infected with type 5 human adenovirus (Ad5) [42], proving that ganciclovir overwhelms Ad5 replication in the liver and that it moderates the Ad5 infections outcomes in these animals, when administered prophylactically or therapeutically. Moreover, its mechanism of action involves the direct inhibition of the adenoviral DNA polymerase.

##### Clinical Studies

This laboratory evidence led to some more recent investigations on patients affected by ocular manifestations induced by adenovirus infection. A few papers showed that the use of topical ganciclovir could be of mild efficacy in such conditions. More specifically, in a retrospective study [43], 200 examinations of patients with adenoviral conjunctivitis diagnosis were revised, to determine the efficacy of the ganciclovir on decreasing recovery time, stopping transmission, and minimizing its consequences. Patients within the first 3 days of adenoviral eye infection (AEI) were allocated into two groups: group 1, including 100 subjects, treated with artificial tears, and group 2 with other 100 persons, using topical ganciclovir along with artificial tears. The study showed a trend toward faster recovery, reduced corneal and conjunctival involvement, and fewer diffusion to the contralateral eye and environment in patients treated with ganciclovir. 

In a double blind, interventional and randomized clinical trial [44], 33 individuals with adenoviral conjunctivitis who had symptoms for five days or fewer were randomly divided in two groups: group 1 (treatment) with 19 subjects using ganciclovir gel and group 2 (control) with 14 individuals treated with preservative free artificial tears. Group 1 showed a not statistically significant tendency of better and quicker answer in terms of signs and symptoms compared with group 2.

In conclusion, a more extensive literature is needed to truly understand if ganciclovir could be effectively used as a therapeutic weapon in adenovirus-induced conjunctivitis. 

#### 3.3.2. Cidofovir

Cidofovir (CDV) is an antiviral agent that belongs to the family of acyclic nucleoside phosphonates (ANPs). The ANPs have been proven to move into the cell via an endocytosis-like mechanism and to be transformed intracellularly by cellular enzymes to their diphosphate metabolites [45].

These enzymes interact with the usual substrates as competitive inhibitors/alternative substrates [46]. In 1996, the intravenous preparation of CDV was approved for the treatment of human cytomegalovirus (HCMV) retinitis in AIDS patients. This medication, however, is usually used off-label to treat severe cases of adenovirus infection [47]. Topical CDV has been studied over the past years to find out whether its use could be of any efficacy in adenovirus ocular manifestations.

##### Laboratory Studies

A beneficial activity of cidofovir on cells and corneal tissues infected by different types of adenoviruses was established in several in vitro and in vivo studies involving animal models [47,48,49,50,51,52,53,54]. In one study [48], the antiviral effectiveness of topical 0.1% (S)-1-(3-hydroxy-2-phosphonylmethoxypropyl)cytosine in the New Zealand rabbit eye model was analyzed. Following topical and intrastromal inoculation with 100 microliters of adenovirus type 5, infected eyes underwent pretreatment (six times daily) beginning one day before inoculation and lasting four days. When matched to the control eyes, the tested eyes had significantly lower ocular viral peak titers and a reduction in the duration of viral shedding. Another study [49] looked at the cidofovir antiviral effect against ocular adenoviral serotypes in vitro, as well as the healing success and eye toxicity of topical therapy on established adenovirus type 5 infection in a rabbit ocular model. In vivo inhibitory results were assessed using serial ocular titers and the length of viral shedding. Slit lamp, together with external evaluation, were performed to assess the presence of ocular toxicity. Even in this case, a reduction in viral titers and viral shedding was noticed, whereas no significant clinical toxicity was shown. 

Cidofovir’s antiviral effects on ocular adenovirus infection were also examined in cotton rat eyes [50], which received intrastromal injection and topical infection by four adenovirus serotypes 4, 5, 8, and 37, and were topically treated with 1% cidofovir eye drops two times per day. For adenovirus serotypes 4, 8, and 37, no proof of prolonged virus replication in cotton rat eyes was observed. Specimens from cidofovir-treated eyes contaminated with adenovirus 5 showed a statistically significant decrease in the mean virus titer and virus shedding length, in comparison to the control group. Because adenovirus ocular infections continue to be a major public health concern due to their high frequency, a viable prophylactic to prevent infection transmission and reduce the occurrence of community epidemics is a critical aim. The effects of cidofovir as a preventive medication were studied in the eyes of New Zealand rabbits infected with adenovirus serotype 5 [51]. This study used 21 rabbits each trial. Two days before viral injection, rabbits were randomly allocated to one of three topical ocular therapy groups, and bilateral treatment began two times per day: (1) 1% cidofovir, (2) 0.5% cidofovir, and (3) the control group. Antiviral prophylaxis with 1% and 0.5% cidofovir was demonstrated to significantly decrease adenovirus type 5 ocular titers, and that it impacted positively on several other parameters, such as positive cultures/total. The New Zealand rabbit ocular model was also utilized to assess the effectiveness of topical 0.5% cidofovir twice daily for 7 days on the replication of several adenovirus serotypes of subgroup C, such as 1, 5 and 6. In one study [52], 20 rabbits (Ad5) were infected topically in both eyes with 1.2 x 10^5^ pfu/eye of the relevant virus (Ad1 and Ad6). After 24 h, the rabbits in each serotype group were randomly assigned to one of two topical treatment groups: (1) 0.5% cidofovir, and (2) control group. For seven days, the treatment was administered twice daily. Treatment with 0.5% cidofovir lowered viral titers on different days, as well as the number of adenovirus positive eyes/total and adenovirus shedding in all types. Nonetheless, in the New Zealand white rabbit ocular model, Ad5 variants R1, R2, and R3 were shown to be resistant to topical therapy with 0.5% cidofovir [53,55].

##### Clinical Studies

Gordon et al. [56] used topical cidofovir 0.2% to treat a 31-year-old patient with confirmed adenoviral conjunctivitis. Symptoms improved significantly after four days, all clinical abnormalities were eliminated by seven days, and the cornea remained clean, while the second eye that had received preventive therapy remained symptom-free.

Because preclinical investigations and phase 1 and 2 clinical trials [24,25,26,27,28,29] indicated that cidofovir had a high potential for use in adenoviral ocular infections, a randomized controlled clinical study [57], that assessed the effectiveness of cidofovir 1% eyedrops with and without cyclosporin A 1% eyedrops in the treatment of acute adenoviral keratoconjunctivitis (AKC) was performed. Thirty-four patients with recent onset acute adenoviral keratoconjunctivitis were randomly assigned to one of four therapy groups: (1) cidofovir four times daily, (2) cidofovir ten times daily, (3) cidofovir plus cyclosporin A, both four times daily, and (4) sodium chloride four times daily (control). Adenoviral PCR from conjunctival samples validated the diagnosis. The therapy lasted 21 days. To assess the degree of conjunctival injection, conjunctival chemosis, punctate epithelial keratitis throughout therapy, and the existence and severity of corneal subepithelial infiltrates, a clinical score was used. Time needed to recovery was also assessed. The results showed that topical cidofovir at 1% concentration is a well-tolerated medicine that causes no discomfort but has no statistically meaningful effect on the course of acute adenoviral keratoconjunctivitis. Cidofovir 1% and cyclosporin A 1%, in particular, had no influence on the number of corneal infiltrates at the end of the 21-day treatment period. The current study’s fivefold concentration of cidofovir, on the other hand, appears to be advantageous in the prevention of severe corneal opacities. Only topical immunosuppressants like prednisolone or cyclosporin have been demonstrated to be useful in treating existing corneal opacities. Their mechanism of action includes the reduction of the immune response to viral antigens that remain in the cornea. Cidofovir, on the other hand, targets the underlying cause of corneal opacities as an antiviral medication. As contrast to steroid therapy, or even less so topical cyclosporine, it may thereby avoid their onset. 

Another randomized, controlled, double-masked study [58] on 39 patients with acute adenoviral keratoconjunctivitis used topical cidofovir and cyclosporine as local treatments. In this study, patients were placed into four therapy groups: (1) cidofovir 0.2% eyedrops, (2) cyclosporine 1% eyedrops, (3) cidofovir 0.2% + cyclosporine 1% eyedrops, and (4) sodium chloride (control). A PCR was used to obtain a diagnosis, while treatment lasted 21 days. The primary considered outcomes were conjunctival hyperemia and chemosis, as well as the presence of superficial punctate keratitis and corneal stromal infiltrates. Surprisingly, subjective improvement in local symptoms occurred only in the cyclosporine group.

#### 3.3.3. Trifluridine

Trifluridine, also known as trifluorothymidine, is a fluorinated pyrimidine nucleoside and a structural analogue of the deoxyribonucleoside, thymidine, as well as the established antiviral agent, idoxuridine [59].

##### Laboratory Studies

Regarding the in vitro research on trifluridine, cell cultures were used to test the capacity to inhibit reference strains of adenoviruses, type 8, type 19, and type 13 [60].

Drug-treated cell cultures experienced less cytopathic effect after infection with all three serotypes, in comparison to untreated, virus-infected cell cultures. In the drug-treated cell cultures, the amount of virus was reduced by over ten-fold for type 8, over 1000-fold for type 19, and 5000-fold for the type 13 isolates.

##### Clinical Studies

A clinical study [61] tried to determine whether trifluridine could be an effective treatment for conjunctivitis due to type 3 adenovirus. Two drops of adenovirus were instilled in the lower conjunctival sac of both eyes of the 30 subjects. All subjects became clinically diseased after 3 days from the instillation. Methisazone suspension was given orally to 10 subjects in the schedule of 4 gm b.i.d. for 3 days, then 4 gm daily for 3 days beginning immediately after infection. Trifluorothymidine 1% eye drops were instilled in the conjunctival sacs of both eyes of 10 other subjects in the dosage of one drop into each eye five times daily for 8 days beginning on the day of infection. Each subject was examined daily for 14 days and twice a week for another 14 days. There was no apparent shortening of the duration of the disease in those taking methisazone or trifluorothymidine and no flare-up of disease occurred when these drugs were stopped. 

However, when 1% trifluridine was used clinically on 21 subjects during an outbreak of keratoconjunctivitis [62] due to a different serotype of adenovirus (type 19), severe and persistent symptoms in 12 patients were completely cleared within 4 days of a 7-day course. 

During an epidemic keratoconjunctivitis caused by adenoviruses, a prospective, double-masked clinical trial [63] of trifluridine, dexamethasone and artificial tears aimed to investigate any potential benefit of the abovementioned drugs in the treatment of these patients. The participants were sorted into three different treatment groups; 25 received 1.0% topic trifluridine, 25 were treated with 0.5% dexamethasone sodium phosphate, while artificial tears were chosen as placebo for the last group. Controls were performed every 5 days during the symptomatic disease, at the time of resolution and 2 weeks after completing the therapy; viral cultures were also performed in all participants. There was no statistical difference between the three treatment groups in reducing the length of signs and symptoms.

Anecdotal use of trifluridine [64] is also described as a potential therapeutic weapon against long term corneal alterations due to adenoviral keratoconjunctivitis, such as corneal subepithelial infiltrates. Five patients with corneal subepithelial infiltrates persevering for roughly a year after epidemic keratoconjunctivitis and one patient with infiltrates persisting for about six years were treated with trifluridine. Earlier topical corticosteroid treatment for the infiltrates was unsuccessful in all patients. After receiving the trifluridine therapy for one week, some subepithelial infiltrates vanished and the opacity of the remaining infiltrates significantly diminished.

#### 3.3.4. Filociclovir

Filociclovir (FCV; also known as cyclopropavir or MBX-400) is a methylenecyclopropane nucleoside analog with broad-spectrum antiviral activity, whose targets include HCMV, VZV, EBV, HHV-6A, HHV-6B, HHV-7 and HHV-8 [65,66,67]. It has successfully completed Human Phase I safety studies [65,68,69].

##### Laboratory Studies

Recently, FCV proved to be a potent inhibitor of Adenovirus in cell cultures [70,71]. In one report, FCV was showed to be a potent and selective inhibitor of 5 HAdV types (4–8) [72]. More specifically, FCV at a dose of 10 mg/kg administered orally, daily, starting from the day before virus challenge, until the end of the experiment, successfully prevented mortality and lowered morbidity in immunosuppressed Syrian hamsters. Furthermore, FCV suppressed liver pathology by inhibiting virus replication in the livers of the infected animals. Another study [73] tried to investigate its antiviral activity against ocular isolates of HAdV, its ocular tolerability, and antiviral efficacy in animal models, concluding that FCV possesses antiviral in vitro activity against a different ocular HAdV species. Its lack of toxicity, good tolerability, and its specific activity against in the Ad5/NZW rabbit ocular model were also proved. 

##### Clinical Studies

Based on these in vitro and in vivo profiles, FCV was selected for clinical tests in humans. Phase IA and IB were conducted; no serious adverse effects were observed, and oral doses as low as 100 mg daily achieved plasma concentrations that are sufficient to inhibit HAdVs in vitro [68,69]. Considering the above-mentioned achievements, both in vitro and in vivo on animal models, filociclovir is a very good candidate for phase II clinical trials in humans, and, more generally, a promising agent against HdAV systemic and ocular infections.

### 3.4. Other Agents

Other non-nucleoside agents have shown antiviral effect on adenovirus, among them the sulfated sialyl lipid NMSO3 [74,75] whose anti-AdV activity consist of inhibition of virus adsorption to cells by NMSO3 binding to viral particles. The endogenous microbicide N-chlorotaurine, whose mechanism is supposed to be the oxidization of thiols and amines, has shown antimicrobial activity against adenoviruses, in addition to bacteria, fungi and HSV-1 [76,77]. N-chlorotaurine has revealed its safety if locally administered in human as well animals [78,79] and proved itself to be more effective than gentamicin in a small phase II study [80]. The non-nucleoside doxovir, also known as CTC-96, is an imidazole derivative, that belongs to a novel class of antiviral drugs, called cobalt chelates, that work by tightly adhering to histidine. Doxovir 50 mg/mL eradicated the virus by day 10 in the Ad5 rabbit replication model, in comparison to day 21 in placebo-treated controls [81]. Even if some more mixtures have shown anti-adenoviral action in vitro, the majority still need to be tested in animal models. Traditional plant-derived substances are among them [82], as well as plant green tea catechins [83] cycloferon [84], lactoferrin [85], heterocyclic Schiff bases of aminohydroxyguanidine tosylate [86,87] a topoisomerase inhibitor [88] and papain and protease inhibitors [89,90], peptidomimetic integrin-binding antagonists [91], human a-defensin peptides [92].

## 4. Discussion

At present, although HAdV is a critical pathogen with a great clinical effect on immunocompromised people, no vaccines or specific antiviral drugs are available to deal with such a disease. As for the growth of specific anti-HAdV remedies, we should consider that serotyping for HAdV is not regularly performed in diagnostic units and that different HAdV serotypes can be the causative vehicle for the same pathology. Considering this, possible antiAdV medications must include antivirals with a broad serotype-spectrum to HAdV; to this end, multiple specific HAdV processes or drug targets, such as entry, DNA replication, or virion assembly, could address the growth of antiviral drugs [92].

As regards to the clinical manifestation of adenovirus on ocular surface, we can identify a spectrum of pathologies with different characteristics. While follicular conjunctivitis (FC) and pharyngeal conjunctivitis fever (PCF) are usually mild conditions that do not have a long-term impact on visual acuity, epidemic keratoconjunctivits (EKC) may have serious consequences on visual acuity. In such manifestation of ocular adenovirus, the acute phase including viral replication generally persists up to 14 days. However, after the EKC acute phase, the corneal stroma is frequently involved by an immune T cell infiltration, resulting in the presence of multiple little, white spots (sub-epithelial infiltrates) that can impair vision (e.g., decreased visual acuity, photophobia) for periods of months or even years [93]. 

To date, there is no approved antiviral drug for adenoviral conjunctivitis, whose management is therefore still based on hygienic and supportive measures. Considering the epidemiological importance of this condition, as well as the above-mentioned possibility of long-term consequences, an effective antiviral treatment would be very important (Table 1).

The design of large, randomized, controlled clinical trials investigating the efficacy of antiviral compounds on human adenoviruses causing ocular disease meets several hurdles in the practice. First, FC and PCF are not referred to physicians in many cases, as these conditions are usually seasonal, self-limited and of short duration. On the other side, EKC does not frequently cause epidemics in the Western world, while it can be considered endemic in less developed countries. Additionally, ocular adenoviruses are usually related to symptoms of low specificity, that can easily mimic those caused by other infectious agents of even by non-infectious ones. The diagnosis of adenoviral conjunctivitis is based on clinical appearance alone in most of cases, thus a specific serotype target is not usually identified. If a restricted serotype breadth of efficacy antiviral is to be tested in clinical trials, quick diagnostic serotyping may also be crucial. In fact, the lack of rapid diagnostic tests in most countries reflects the challenge of early identification in the clinic, in most countries. An effective clinical trial should also focus on the logistics and costs of fast laboratory diagnosis (real-time PCR, enzyme immunoassay, culture) for appropriate enrolment. 

Therefore, when correct diagnosis is delayed, the efficacy of antiviral drugs is more difficult to be proven. 

## 5. Conclusions

Many compounds with antiviral activity on human adenovirus serotypes have been analyzed over the years to find an effective antiviral drug to treat adenoviral ocular manifestations. Some of them have showed efficacy in both in vitro tests and in pre-clinical trials but were not proved to be reliable in further investigations. Nevertheless, some of them are promising, but large, randomized, controlled clinical trials are needed to confirm their reliability in the treatment of ocular adenovirus conjunctivitis, even if the design of such trials meets several problems due to the specific characteristics of adenovirus-induced ocular diseases. More specifically, based on the results of our research, we can identify cidofovir, filociclovir, and ganciclovir as the most promising agents for topical use against ocular adenovirus; nevertheless, some new agents (NMS03 and CTC-96, among others) have proved to be useful against adenovirus, but large, prospected, randomized clinical trials are needed confirm their efficacy and safety. As for the other agents included in this review, some of them (zalcitabine) could be worth further analysis on human beings in order to prove their efficacy on human adenovirus and its ocular manifestations, as they showed good results in in vitro models, but were not tested on animals or in clinical trials. Finally, a better comprehension of the mechanisms used by human adenoviruses to colonize and inflame ocular surface could help to find a more targeted, safe, and cost-effective antiviral compound to better manage sight-threatening, adenovirus-induced ocular manifestations.

## Figures and Tables

**Table 1 microorganisms-10-02014-t001:** Comparison of different compounds.

		IC50	CC50
Acyclovir	Herpesvirus DNA polymerase inhibition, integration into and ending of the developing viral DNA chain, and inactivating the viral DNA polymerase	n.a.	n.a.
Ganciclovir	Viral DNA polymerase inhibition, viral DNA replication suppression by ganciclovir-5′-triphosphate (ganciclovir-TP)	n.a.	212 mg/mL (827 mM) ^40^
Cidofovir	Enters cell via an endocytosis-like mechanism and is transformed intracellularly by cellular enzymes to their diphosphate metabolites. These enzymes interact with the usual substrates as competitive inhibitors/alternative substrates	0.018 to 5.47 mg/mL ^54^ 0.487 to 30.304 µM ^71^	0.49 to 30.3 µM ^73^
Ribavirin	Quickly phosphorylated by intracellular enzymes, the triphosphate inhibits influenza virus RNA polymerase activity and competitively prevents the guanosine triphosphate–dependent 5′ capping of influenza viral messenger RNA	396 to >500 µM 48 to 108 µM ^18^	n.a.
Zalcitabine	Two antiretroviral mechanisms have been proposed: integration of dideoxynucleoside triphosphates onto expanding strands of viral DNA or competition for reverse transcriptase with endogenous nucleoside triphosphates	3.5 to 48.7 mg/mL ^25^	4282 μg/mL (20.3 mM) ^24^
Vidarabine	Upon cellular uptake, this nucleoside analogue is physiologically triggered by 5′-triphosphorylation. Then, it inhibits intracellular enzymes and/or delays or terminates nucleic acid production, as they are integrated into nascent DNA and RNA strands	n.a.	n.a.
Idoxuridine	Effective reversible inhibitor of the enzyme thymidilate synthetase, which converts irridine monophosphate to thymidine monophosphate. DNA synthesis is decreased because of this thymidilate synthetase inhibition	n.a.	n.a.
Trifluridine	Fluorinated pyrimidine nucleoside and a structural analogue of the deoxyribonucleoside, thymidine, as well as the established antiviral agent, idoxuridine	n.a.	n.a.
Filociclovir	Methylenecyclopropane nucleoside analog with broad-spectrum antiviral activity	0.496 to 4.684 µM ^71^	0.50 to 4.68 µM ^73^

^71^: Ad3, Ad4, Ad5, Ad7a, Ad8, Ad19/64 and Ad37. ^40^: HAdV3 (species B), HAdV4 (species E), and 8, 19a, and 37 (species D). ^56^: Ad5. ^73^: HAdV5. ^18^: Serotypes from species A, B, D, E and F (resistant); serotypes from species C (non resistant). ^24^: A549. n.a. = not applicable.

## Data Availability

The data presented in this study are fully available in the text.

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
