# Peer review of "Antiviral Drugs in Adenovirus-Induced Keratoconjunctivitis"

_microorganisms, 2022, doi:10.3390/microorganisms10102014_

Round 1

Reviewer 1 Report

This review by Imparato et al. provides a comprehensive overview of potential agents for the management of adenovirus keratoconjunctivitis. The scope is impressive, but several aspects warrant attention:

1. The grammar and English require extensive revision. In many parts of the manuscript the text is entirely unintelligible, often with incomplete sentence fragments as well as duplicated sentences.

2. An evaluation of the quality of the clinical studies included is crucial to understand how heavily to weight the results of said studies. There is no mention of potential sources of bias, confounding, un-masking, or other potential flaws in the studies described. Further, the level of evidence of different studies is entirely ignored. Please discuss the limitations of each study as you present it.

3. The organization of the manuscript should be improved. For each medication, please provide the same sub-sections (e.g. "laboratory studies" to review the in vitro/in vivo studies, and "clinical studies" to highlight case reports/series, observational studies, and interventional studies). Otherwise it is unclear whether some drugs simply have no clinical data, or of the authors simply chose to omit them. Zalcitabine is a good example of this issue. It would also be helpful to begin each section with lower quality evidence, and end with the highest quality evidence. This would avoid issues such as in the trifluridine clinical studies section, where RCTs (which showed negative results) are described first, and the section is concluded with a case series of 5 patients suggesting a positive effect. This gives the false impression that trifluridine has clinical evidence to support its use.

4. Were "adenoviral conjunctivitis" and "antiviral" really the only search terms used? If so this is problematic, as you would likely miss studies referring to "epidemic keratoconjunctivitis", "pink eye", "viral conjunctivitis", etc. Please clarify if these really were the only search terms used, and if so provide some justification for why.

5. The conclusions are often not supported by the evidence presented. For example, the authors state that the RCT conducted with ganciclovir "showed a beneficial effect of topical ganciclovir", but 2 sentences later they say "in conclusion, group 1 (the ganciclovir group) showed a not statistically significant tendency of better and quicker answer in terms of signs and symptoms compared with group 2." A non-significant "trend" is not a beneficial effect. There is a similar example in the cidofovir section. "the results showed that topical cidofovir at 1% concentration is a well-tolerated medication that causes no discomfort but has no statistically meaningful effect on the course of acute adenoviral conjunctivitis." Then 2 sentences later, the authors state "the current study's fivefold concentration of cidofovir, on the other hand, appears to be advantageous in the prevention of severe corneal opacities." These sentences are contradictory. This emphasizes the earlier point about discussing the level of evidence and strength of any effects observed in these clinical studies. 

6. The outcomes used for the clinical studies described are highly variable. the authors rightly indicate that additional trials are needed, but please provide suggestions for what clinical outcomes should be evaluated to facilitate design of these studies to ensure the correct questions are answered. 

Minor Comments:

1. The first sentence of the fourth paragraph of the Introduction states that EKC is a more common manifestation of HAdV infection than pharyjgoconjunctival fever. This is not even close to accurate, please revise.

2. Please define all acronyms before use (e.g. "AEI" in the ganciclovir clinical trials section).

3. In one of the trifluridine studies, the authors state that "two drops of adenovirus were instilled in the lower conjunctival sac of both eyes of the 30 subjects," and "all subjects became clinically diseased after 3 days..." If this is accurate, the ethics of this study seem highly dubious. Unless IRB approval was obtained, I would recommend not including a potentially unethical study in your review.

4. The first sentence of the Discussion states that "HAdV is a valuable pathogen with great clinical influence in immunocompromised people". I'm not sure what this is trying to say, but it clearly needs to be rephrased.

Author Response

  1. The grammar and English require extensive revision. In many parts of the manuscript the text is entirely unintelligible, often with incomplete sentence fragments as well as duplicated sentences.

Thank you, We revised the English 

  1. An evaluation of the quality of the clinical studies included is crucial to understand how heavily to weight the results of said studies. There is no mention of potential sources of bias, confounding, un-masking, or other potential flaws in the studies described. Further, the level of evidence of different studies is entirely ignored. Please discuss the limitations of each study as you present it.

Thank you for your comment, we summarized this issue in the discussion section

  1. The organization of the manuscript should be improved. For each medication, please provide the same sub-sections (e.g. "laboratory studies" to review the in vitro/in vivo studies, and "clinical studies" to highlight case reports/series, observational studies, and interventional studies). Otherwise, it is unclear whether some drugs simply have no clinical data, or of the authors simply chose to omit them. Zalcitabine is a good example of this issue. It would also be helpful to begin each section with lower quality evidence, and end with the highest quality evidence. This would avoid issues such as in the trifluridine clinical studies section, where RCTs (which showed negative results) are described first, and the section is concluded with a case series of 5 patients suggesting a positive effect. This gives the false impression that trifluridine has clinical evidence to support its use.

Thank you for your comment, we modified according to your suggestions

  1. Were "adenoviral conjunctivitis" and "antiviral" really the only search terms used? If so this is problematic, as you would likely miss studies referring to "epidemic keratoconjunctivitis", "pink eye", "viral conjunctivitis", etc. Please clarify if these really were the only search terms used, and if so provide some justification for why.

Thank you for your comment. The research was conducted with such terms as the terms "adenoviral conjunctivitis" generally indicates the entire spectrum of ocular nosographic entities attributable to adenoviruses (such as epidemic conjunctivitis). The terms "viral conjunctivitis " could be too general, as the aim of the study was to investigate the activity of drugs specifically on adenovirus-induced manifestations, and no other viral agents. Finally, the terms "pink eye" could have been confusing, as it includes numerous eye diseases, (e.g. glaucoma, keratitis, ocular foreign body…) of which only a small part is attributable to adenoviral conjunctivitis.

  1. The conclusions are often not supported by the evidence presented. For example, the authors state that the RCT conducted with ganciclovir "showed a beneficial effect of topical ganciclovir", but 2 sentences later they say "in conclusion, group 1 (the ganciclovir group) showed a not statistically significant tendency of better and quicker answer in terms of signs and symptoms compared with group 2." A non-significant "trend" is not a beneficial effect.

 Thank you for your comment, we revised the sentence (page 6, lines 254-260)

There is a similar example in the cidofovir section. "the results showed that topical cidofovir at 1% concentration is a well-tolerated medication that causes no discomfort but has no statistically meaningful effect on the course of acute adenoviral conjunctivitis." Then 2 sentences later, the authors state "the current study's fivefold concentration of cidofovir, on the other hand, appears to be advantageous in the prevention of severe corneal opacities." These sentences are contradictory. This emphasizes the earlier point about discussing the level of evidence and strength of any effects observed in these clinical studies. 

Thank you for your comment, but in our opinion the two sentences are not in contradiction. In fact, cidofovir has no statistically meaningful effect on the course of acute adenoviral conjunctivitis, but appears to be advantageous in the prevention of severe corneal opacities (page 7, lines 344-350)

Minor Comments:

  1. The first sentence of the fourth paragraph of the Introduction states that EKC is a more common manifestation of HAdV infection than pharyjgoconjunctival fever. This is not even close to accurate, please revise.

Thank you for your comment, you are right, the word “ocular” was missing, we added it. (page 1, line 39)

  1. Please define all acronyms before use (e.g. "AEI" in the ganciclovir clinical trials section).

Thank you, we added the terms before acronyms (page 6 line 248)

  1. In one of the trifluridine studies, the authors state that "two drops of adenovirus were instilled in the lower conjunctival sac of both eyes of the 30 subjects," and "all subjects became clinically diseased after 3 days..." If this is accurate, the ethics of this study seem highly dubious. Unless IRB approval was obtained, I would recommend not including a potentially unethical study in your review.

Thank you for your comment. The study was conducted in 1968 on prisoners, at that time no IRB was required for such studies. Even if we agree that this is unethical, we cannot forget the results of all the experiments conducted in the past.

  1. The first sentence of the Discussion states that "HAdV is a valuable pathogen with great clinical influence in immunocompromised people". I'm not sure what this is trying to say, but it clearly needs to be rephrased.

Thank you for your comment, we rephrased the sentence (page 10 lines 477-479)

Reviewer 2 Report

Imparato et al. is an attempt to review the literature on antiviral drugs that could have potential in treating adenovirus-induced keratoconjunctivitis. Given the lack of approved specific therapies, several antivirals are used on an off-label basis. This review could be of benefit to clinicians. However, I have the following major criticisms:

The authors have listed all antiviral drugs reported in the literature to have been tested in vitro oe clinically. As correctly stated in the conclusions section, some of them showed promise, while other didn't seem to be beneficial. Nonetheless, the authors treated them more or less equally in their manuscript. I would suggest including 2 major subsections: one for the compounds that showed potency in vitro, in vivo and possibly clinically, and a second subsection on the compounds with no observed or questionable potency. For example, acyclovir, vidarabine and idoxuridine didn't seem to be potent against adenoviruses.

This manuscript badly needs a summary table that compares the compounds side by side. This table should also include information on the in vitro potency and selectivity (IC50 and CC50), which was completely lacking for several compounds described in the manuscript.

The authors have completely ignored Filociclovir (FCV), one of the most promising compounds with proven in vitro and in vivo activity against human adenoviruses (please check references PMID: 33810229, PMID: 32816736 and PMID: 31940473). Filociclovir has been shown to be safe clinically in phase I trials (PMID: 32134483 and PMID: 31285228), and is currently being developed for treating ocular human adenovirus infections. I suggest including a whole new section to summarize the published literature on this compound to give a more accurate picture of anti-adenovirus drug landscape to the reader.

In the conclusions sections, the authors stated unjustifiably that ganciclovir (GCV) is the most promising agent for topical use against ocular adenovirus infections. When compared to CDV and FCV, GCV was the least potent in vitro with IC50 values of about 60 uM (PMID: 30287226). Furthermore, both GCV and CDV are known to be toxic. I suggest deleting or modifying this sentence to reflect compounds of proven promise based on published data.

The English grammar and spelling needs careful revision.

Other minor comments:

The last paragraph of the discussion section, which highlights the challenges facing clinical testing of adenovirus inhibitors is excellent. I suggest highlighting the lack of rapid diagnostic tests in most countries, except Japan. Adenovirus infection is also notifiable only in Japan, which reflects the challenge of early identification in the clinic, especially in Western countries. In particular, this paragraph deserves more elaboration because it captures one of the most important hurdles in anti-adenovirus drug development.

The word "reproduction" is not accurate when used in the context of viruses. I suggest using "replication" instead.

Deoxyribonucleic acid can be safely abbreviated to DNA.

What do you mean by "clinical gravity"?

The last sentence in vidarabine clinical trials is very confusing. Please, rephrase.

Please, replace "anecdotic" with "anecdotal".

Author Response

-The authors have listed all antiviral drugs reported in the literature to have been tested in vitro oe clinically. As correctly stated in the conclusions section, some of them showed promise, while other didn't seem to be beneficial. Nonetheless, the authors treated them more or less equally in their manuscript. I would suggest including 2 major subsections: one for the compounds that showed potency in vitro, in vivo and possibly clinically, and a second subsection on the compounds with no observed or questionable potency. For example, acyclovir, vidarabine and idoxuridine didn't seem to be potent against adenoviruses.

Thank you for your comment, we modified the text accordingly

-This manuscript badly needs a summary table that compares the compounds side by side. This table should also include information on the in vitro potency and selectivity (IC50 and CC50), which was completely lacking for several compounds described in the manuscript.

We added the table, providing IC50 and CC50, when available

-The authors have completely ignored Filociclovir (FCV), one of the most promising compounds with proven in vitro and in vivo activity against human adenoviruses (please check references PMID: 33810229, PMID: 32816736 and PMID: 31940473). Filociclovir has been shown to be safe clinically in phase I trials (PMID: 32134483 and PMID: 31285228), and is currently being developed for treating ocular human adenovirus infections. I suggest including a whole new section to summarize the published literature on this compound to give a more accurate picture of anti-adenovirus drug landscape to the reader.

Thank you for your comment, we added a section describing studies on Filociclovir (pages 9-10, lines 432-456)

-In the conclusions sections, the authors stated unjustifiably that ganciclovir (GCV) is the most promising agent for topical use against ocular adenovirus infections. When compared to CDV and FCV, GCV was the least potent in vitro with IC50 values of about 60 uM (PMID: 30287226). Furthermore, both GCV and CDV are known to be toxic. I suggest deleting or modifying this sentence to reflect compounds of proven promise based on published data.

Thank you for your comment, we modified the conclusions (page 11, lines 527)

The English grammar and spelling needs careful revision.

We revised the English

Other minor comments:

The last paragraph of the discussion section, which highlights the challenges facing clinical testing of adenovirus inhibitors is excellent. I suggest highlighting the lack of rapid diagnostic tests in most countries, except Japan. Adenovirus infection is also notifiable only in Japan, which reflects the challenge of early identification in the clinic, especially in Western countries. In particular, this paragraph deserves more elaboration because it captures one of the most important hurdles in anti-adenovirus drug development.

Thank you for your comment, We added this concept in the discussion (page 11, lines 511-513)

The word "reproduction" is not accurate when used in the context of viruses. I suggest using "replication" instead.

We made the changes

Deoxyribonucleic acid can be safely abbreviated to DNA.

We made the changes

What do you mean by "clinical gravity"?

Sorry, we meant “severity” (page 5, line 199)

The last sentence in vidarabine clinical trials is very confusing. Please, rephrase.

We rephrased, thank you (page 4, lines 161-163)

Please, replace "anecdotic" with "anecdotal".

We made the change (page 9, line 423 )

Reviewer 3 Report

Title: Antiviral Drugs in adenovirus-induced keratoconjunctivitis

·       Summary

In this manuscript, the authors reported various antiviral drugs in adenovirus-induced keratoconjunctivitis. In this review, the authors well-organized previously published papers and well-explained for each drug. However, the manuscript has to carefully consider reviewing next minor issues below.

·       Major issues

1.      Check “space”. In some place, the authors added double space and add no space.

2.      Use citation program such as Endnote. It looks the authors added reference numbers by manually. The authors added “].” after [5] in page 2. Also, there is no “[ ]” for 78,79 in page 5. Check whole manuscript.

3.      Unify unit. In some places, the authors used “%” but “percent” in other places.

4.      Unify format in whole manuscript. 36 patients or thirty-six patients.

5.      Change “0,5%” to “0.5%”. Check whole manuscript.

6.      Check other mistakes.

Author Response

Major issues

  1. Check “space”. In some place, the authors added double space and add no space.
  2. Use citation program such as Endnote. It looks the authors added reference numbers by manually. The authors added “].” after [5] in page 2. Also, there is no “[ ]” for 78,79 in page 5. Check whole manuscript.
  3. Unify unit. In some places, the authors used “%” but “percent” in other places.
  4. Unify format in whole manuscript. 36 patients or thirty-six patients.
  5. Change “0,5%” to “0.5%”. Check whole manuscript.
  6. Check other mistakes.

We checked the whole manuscript and corrected references, units, and other mistakes.

Round 2

Reviewer 2 Report

The authors have done a great job in improving this manuscript. However, Table 1 is still incomplete. Please, fill in the missing IC50 and CC50 values.

Author Response

The authors have done a great job in improving this manuscript. However, Table 1 is still incomplete. Please, fill in the missing IC50 and CC50 values.

Thank you for your suggestion, we modified the table according to the results provided by the literature